# Metabolite-Mediated Responses of Phyllosphere Microbiota to Rust Infection in Two *Malus* Species

Yunxia Zhang,[a,b] Bin Cao,[c] Yumei Pan,[a,b] Siqi Tao,[a,b] Naili Zhang[a,b]

[a]The Key Laboratory for Silviculture and Conservation of Ministry of Education, Beijing Forestry University, Beijing, People's Republic of China
[b]Ecological Observation and Research Station of Heilongjiang Sanjiang Plain Wetlands, National Forestry and Grassland Administration, Shuangyashan, People's Republic of China
[c]State Key Laboratory of Mycology, Institute of Microbiology, Chinese Academy of Sciences, Beijing, People's Republic of China

**ABSTRACT** Plants recruit beneficial microbes to enhance their ability to fight pathogens. However, the current understanding of microbial recruitment is largely limited to belowground systems (root exudates and the rhizosphere). It remains unclear whether the changes in leaf metabolites induced by infectious pathogens can actively recruit beneficial microbes to mitigate the growth of foliar pathogens. In this study, we integrated microbiome and metabolomic analyses to systematically explore the dynamics of phyllosphere fungal and bacterial communities and key leaf metabolites in two crabapple species (*Malus* sp. "Flame" and *Malus* sp. "Kelsey") at six stages following infection with *Gymnosporangium yamadae*. Our results showed that the phyllosphere microbiome changed during lesion expansion, as highlighted by a reduction in bacterial alpha-diversity and an increase in fungal alpha-diversity; a decreasing and then an increasing complexity of the microbial co-occurrence network was observed in Kelsey and a decreasing complexity occurred in Flame. In addition, nucleotide sugars, diarylheptanoids, and carboxylic acids with aromatic rings were more abundant in early stages of collection, which positively regulated the abundance of bacterial orders *Pseudomonadales* (in Kelsey), *Acidimicrobiales*, *Bacillales*, and *Flavobacteriales* (in Flame). In addition, metabolites such as flavonoids, lignin precursors, terpenoids, coumarins, and quaternary ammonium salts enriched with the expansion of lesions had a positive regulatory effect on fungal families *Rhynchogastremataceae* and *Golubeviaceae* (in Flame) and the bacterial order *Actinomycetales* (in Kelsey). Our findings highlight that plants may also influence phyllosphere microorganisms by adjusting leaf metabolites in response to biotic stress.

**IMPORTANCE** Our findings demonstrate the response patterns of bacterial and fungal communities in the *Malus* phyllosphere to rust fungus *G. yamadae* infection, and they also reveal how the phyllosphere microbiome changes with the expansion of lesions. We identified several metabolites whose relative abundance varied significantly with lesion expansion. Using a framework for assessing the role of leaf metabolites in shaping the phyllosphere microbiome of the two *Malus* species, we identified several specific metabolites that have profoundly selective effects on the microbial community. In conclusion, our study provides new evidence of the ecological niche of the phyllosphere in supporting the "cry for help" strategy for plants.

**KEYWORDS** metabolomic, phyllosphere microbiome, rust infection, *Malus* species

Address correspondence to Siqi Tao, taosq@bjfu.edu.cn, or Naili Zhang, zhangnaili@bjfu.edu.cn.

The authors declare no conflict of interest.

Historically, most studies on plant-microbe interactions have focused on a small set of specific nutrient-colonizing symbionts and pathogens (1, 2). The advent of new sequencing technologies and technical innovations in the cultivation of microbes in the last 2 decades has facilitated our understanding of plant microbiota; more specifically, all nonsterile plants are colonized by a diverse microbiome that is critical for host performance

(3–5), with three major mechanisms: (i) to improve plant nutrients (6), (ii) to suppress pathogen invasion (7), or (iii) to modulate tolerance to abiotic stresses (8).

Despite the substantial potential of the microbiome in improving plant health, there are still major knowledge gaps regarding the strategy by which plants manipulate the associated microbiomes. Over the last few decades, a plethora of studies have documented that a microbiome assembly can be modulated by plant traits such as the plant phenotype, immune responses, and metabolic profile (9). Considering this plant-microbe interaction from the perspective of metabolites, specific plant metabolites clearly have beneficial or antagonistic effects on distinct microbes (10). Several studies have investigated specific metabolites that exert bioactive effects on microbial community assemblies (10–14). A pioneering study deciphered a strategy defined as "cry for help" in which plants may actively recruit beneficial members from the rhizosphere microbiota through producing multiple root exudates to mitigate biotic stresses (8). An infection of *Arabidopsis thaliana* leaves with *Pseudomonas syringae* pathovar *tomato* DC3000 clearly induces malic acid secretion from its roots, thereby promoting the recruitment of *Bacillus subtilis* strain FB17 and triggering induced systemic resistance (15). Similarly, Wen et al. (16) found that foliar pathogen infection of *A. thaliana* released additional long-chain fatty acids and amino acids into the rhizosphere to recruit specific *Pseudomonas* populations. A local infection of cucumber roots by *Fusarium oxysporum* resulted in increased tryptophan and reduced raffinose associated with root colonization via beneficial *Bacillus amyloliquefaciens* (17). Recent investigations of rhizosphere microbiome compositions have revealed that the metabolic interplay between plants and microbiomes involves a wide diversity of plant-derived metabolites (18). In addition to primary metabolites, plant secondary metabolites are vital in recruiting beneficial microorganisms to facilitate plant defense, such as coumarins (12, 19), triterpenes (14), camalexin (20), flavonoids (21), benzoxazinoids (13, 22), and phytohormones (23). Despite considerable progress being made in dissecting the links between specific plant metabolites with distinct microbial responses, researchers have overwhelmingly focused on the belowground system, with the aboveground components being largely overlooked.

As an aboveground component of terrestrial plants, the phyllosphere is an important ecological niche for plants (24, 25) that is inhabited by a variety of microorganisms in both epiphytic (an organism that grows on the surface of a plant) and endophytic (an organism that lives within a plant) niches. Similar to rhizosphere microbiomes, studies have shown that phyllosphere-colonizing microbes are critical in maintaining plant health and suppressing the overgrowth of foliar pathogens; for example, they can suppress plant diseases by competing with pathogens for nutrients or space (26) by direct suppression (e.g., through the production of antibiotics or physical barriers) (27, 28) or by modulating plant defense responses to enhance plant immunity (29). The microbial community in the phyllosphere can also fluctuate in response to pathogen infections (30–32). For example, the phyllosphere microbiome of citrus leaves infected with the melanistic pathogen *Diaporthe citri* showed a marked reduction in bacterial community evenness, a significantly different bacterial community structure, and an increasingly complex microbial network; the specific microbial taxa *Methylobacterium*, *Pantoea*, and *Acinetobacter* were substantially enriched (33). In wheat, the bacterial community richness of susceptible cultivars showed a significant reduction in *Zymoseptoria tritici*-infected leaves compared to that in healthy leaves (34). In addition, reduced diversity or increased abundance of some potentially beneficial microorganisms has also been reported in the leaves of apples (35), *Arabidopsis* (30), tobacco (36, 37), and wheat (38) upon pathogen infection. However, the potential roles of leaf metabolites in assembling phyllospheric microbial communities remain largely unexplored.

Two *Malus* spp.—"Flame" and "Kelsey"—are two ornamental deciduous crabapple trees that are widely used in urban landscaping in China owing to their beautiful colors and graceful shape (39, 40). However, both are susceptible to infection by the rust fungus *Gymnosporangium yamadae* (40). A heavy infection can substantially influence the ornamental value of crabapple trees. *G. yamadae* is a heteroecious and demicyclic rust fungus, which means that it produces four morphologically distinct spores alternating

between two taxonomically different hosts (*Malus* spp. and *Juniperus chinensis*) throughout its life cycle (41, 42). After heavy spring rain, mature telia absorb water and germinate to produce gelatinous tendrils comprising haploid basidiospores that can disperse into the air. Basidiospores parasitize the surface of *Malus* species, and successful infection leads to orange-yellow spots (spermogonia) emerging on the upper surface of the leaves (41). With the development of rust, the spots enlarge significantly, with distinct coloring patterns among various *Malus* species (43). Moreover, metabolite contents in *Malus* leaves significantly changed after an infection by *G. yamadae*, with different *Malus* species exhibiting accumulations of pigment-related flavonoids in rust spots and adjacent tissues (43). In an earlier field experiment, we found distinct purple and orange patches on rust-infected tissues on the leaves of *Malus* spp. Kelsey and Flame, respectively. However, how plant metabolites regulate pigment formation and their fundamental role in regulating phyllosphere microbial communities requires further investigation.

In this study, we conducted a sequential collection of rust infected leaves of Flame and Kelsey, along with diseased lesion expansion. A combined method of Illumina MiSeq amplicon sequencing and liquid chromatography-mass spectrometry analysis was used to systematically identify key metabolites in the leaves that are crucial for host-pathogen-microbiota communication.

We aimed to (i) assess how phyllosphere microbial communities respond to *G. yamadae* infection along with rust spot enlargement, mainly focusing on microbial community diversity, composition and structure, and co-occurrence association; (ii) identify the key metabolites with significantly different relative abundances among stages; and (iii) investigate the regulatory effect of leaf metabolites on shaping phyllosphere assemblages and screen candidate beneficial microbes that are significantly related to the shift of metabolite profiles. Understanding how plants shape their microbiome upon pathogen attack will enable the development of biocontrol strategies using specific signals to recruit specific beneficial microbes from a given environment.

## RESULTS

**Variation in diseased spots of two *Malus* cultivars after rust infection.** Six consecutive samplings of *G. yamadae*-infected crabapple leaves and healthy leaves have been conducted. In total, 36 samples for both *Malus* spp. Flame and Kelsey, with three biological replicates per stage, were collected. We found that diseased leaves had substantially enlarged spots from stages 1 to 6 (Fig. 1a). Moreover, the two cultivars exhibited distinct coloration patterns after rust infection. The lesions of Kelsey transitioned from light purple (early, stages 1 and 2), to purplish red (middle, stages 3 and 4), and finally to dark purple (late, stages 5 and 6); in contrast, the lesions of Flame transitioned from light yellow (early, stages 1 and 2), to dark yellow (middle, stages 3 and 4), and finally to orange (late, stages 5 and 6).

**Metabolic alterations in leaves of two *Malus* cultivars with expansion of diseased lesions.** The compositions of leaf exudates during lesion expansion in *Malus* spp. Flame and Kelsey were analyzed by ultraperformance liquid chromatography-electrospray ionization mass spectrometry. A total of 1,386 metabolite features were detected and clustered into 152 subclasses (see Table S1). Exudates from *G. yamadae*-infected *Malus* leaves comprised a broad range of metabolites, such as sugars, nucleotides, amino acids, fatty acids, coumarins, and flavonoids (see Table S1).

Principal-component analysis (PCA) of the metabolites showed a strong correlation between the biological replicates of each stage, with a clear separation between the independent stages of the two cultivars (PC1 explained 25.9% of the variance in Kelsey and 27.2% in Flame). Specifically, the metabolic profiles of the early stages (stages 1 and 2) were distinct from those of stage 6 (Fig. 1b). In addition, to obtain a global overview of metabolite profiles with lesion enlargement, we performed a correlation heatmap analysis of the metabolite subclasses of the two cultivars. In general, the signal intensity of most subclasses showed a significant increase from the early (stages 1 and 2) to the late stages (stages 5 and 6) (Fig. 1c).

Furthermore, an orthogonal partial least square-discriminant analysis (OPLS-DA) and Wilcox test analysis were applied to identify the differential metabolites, and the

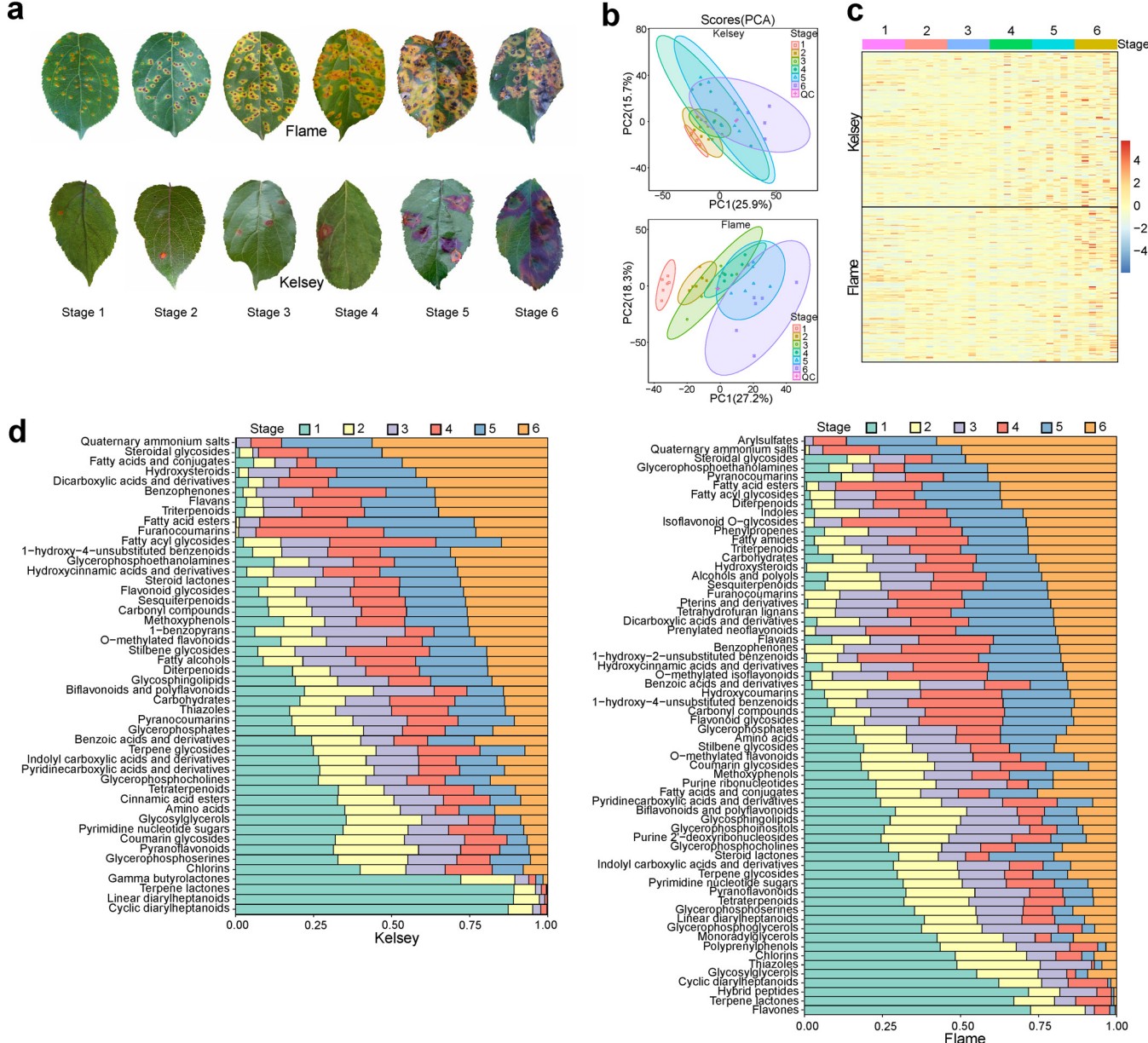

**FIG 1** Overview of variations in metabolites with rust infection. (a) Expansion of rust spots on leaves of *Malus* spp. Flame and Kelsey infected by *G. yamadae*. (b) PCA of the metabolite profiles in rust infected leaves of the two *Malus* cultivars. QC indicates quality control samples that were performed by mixing equal volumes of all samples. (c) Correlation heatmap analysis of the metabolite subclasses in both cultivars. (d) Changes in the abundance of each metabolite subclass during lesion expansion. All abundances were normalized to a range from 0 to 1, with 1 representing the sum of each metabolite released. OPLS-DA, Wilcoxon's test, and the fold change (FC) were used to determine the subclasses with significant differences across expansion of lesions (VIP >1, $P < 0.05$ and FC $\geq$ 2 or $\leq$ 0.5).

integration results are shown in Tables S2 and S3. We determined 48 and 64 subclasses of leaf metabolites whose abundance varied significantly with lesions expansion in Kelsey and Flame, respectively (Fig. 1d). In *G. yamadae*-infected Kelsey leaves, diarylheptanoids were significantly abundant (>80%) at the first stage but almost disappeared after the fourth stage. Notably, the abundance of amino acids, pyrimidine nucleotide sugars, and carboxylic acids with aromatic rings gradually decreased from the first to sixth stage. In contrast, the abundance of flavans, flavonoid glycosides, sesquiterpenoids, triterpenoids, hydroxycinnamic acids, some aromatic compounds, and especially quaternary ammonium salts clearly increased with lesion expansion. In addition, fatty acid esters, fatty acyl glycosides, and furanocoumarins were significantly more abundant in the fourth and fifth stages.

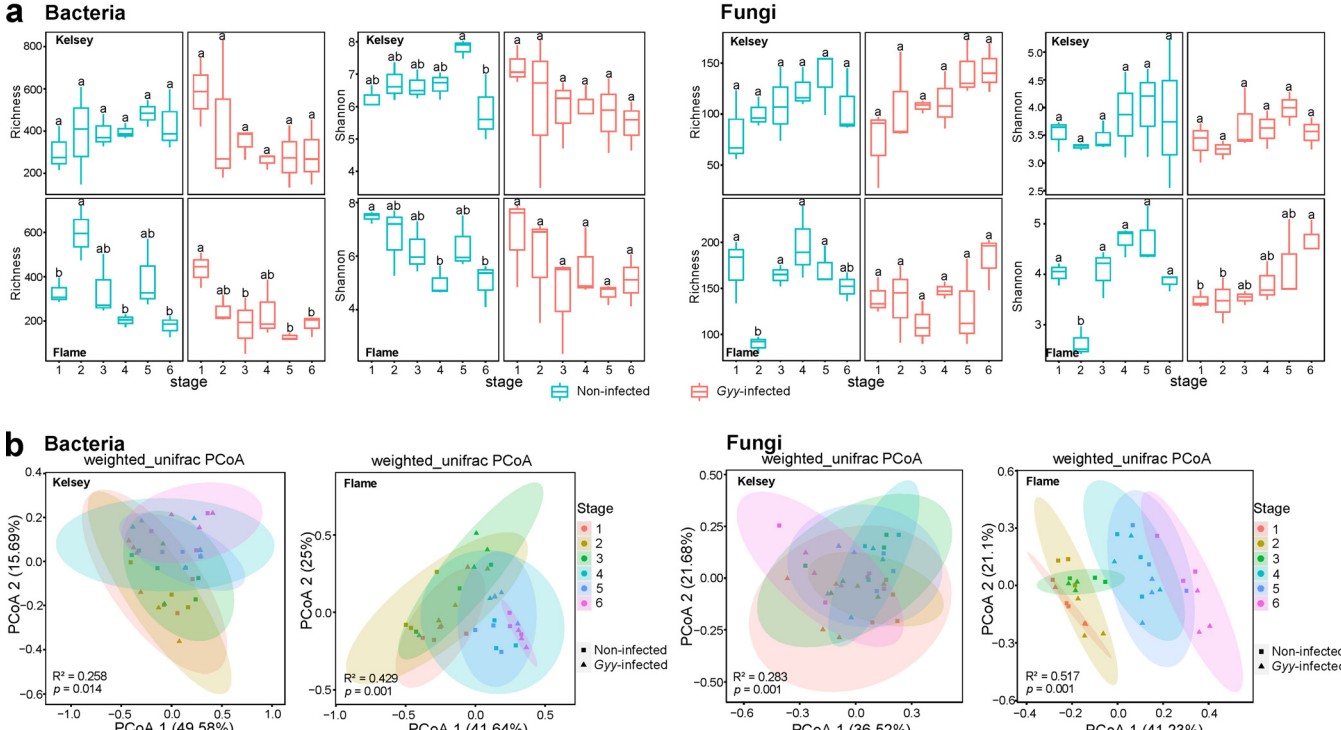

**FIG 2** Shift in the phyllosphere microbiome with rust infection. (a) Changes in alpha-diversity (indicated by Shannon indices and richness indices) of phyllosphere bacterial and fungal communities in noninfected leaves and *G. yamadae*-infected leaves of two *Malus* cultivars (Flame and Kelsey) with six developmental stages of rust disease. Different letters indicate statistically significant differences among stages, as determined by one-way ANOVA with *post hoc* Tukey's HSD test ($P < 0.05$) or Kruskal-Wallis test ($P < 0.05$). Box plots show the range of estimated values between 25 and 75%, and the median, minimum, and maximum observed values within each data set. (b) PCoA of bacterial and fungal communities using the weighted UniFrac distance matrix. Samples were sorted according to sample stages (1 to 6) and leaf conditions (noninfected versus *G. yamadae* infected).

In *G. yamadae*-infected Flame leaves, flavone and hybrid peptides were enriched (more than 60%) in the first stage, but their abundance then decreased sharply (Fig. 1d). Notably, the abundances of diarylheptanoids, purine 2'-deoxyribonucleosides, pyrimidine nucleotide sugars, purine ribonucleotides, and carboxylic acids with aromatic rings decreased gradually from the first to the sixth stage. In contrast, the abundances of sesquiterpenoids, triterpenoids, flavans, isoflavonoid O-glycosides, indoles, diterpenoids, aromatic compounds, furanocoumarins, pyranocoumarins, tetrahydrofuran lignans, and especially quaternary ammonium salts and arylsulfates increased following lesion expansion. In addition, *o*-methylated flavonoids, flavonoid glycosides, and hydroxycoumarins were significantly more abundant in the fourth and fifth stages.

**Diversity and structure of the phyllosphere microbial community shift with lesion expansion.** A total of 4,074,673 16S rRNA and 5,094,401 internal transcribed spacer (ITS) high-quality sequences were obtained via Illumina MiSeq sequencing. The number of sequences varied among samples, with 16S rRNA reads ranging from 31,643 to 112,837 and ITS reads ranging from 50,668 to 111,958 (see Table S4). Quality filtering, denoising, merging, and chimera screening processes resulted in 9,118 bacterial amplicon sequence variants (ASVs) (see Table S5) and 2,759 ITS fungal ASVs. Because the fungal ITS primers also target plant DNA, we found 134 ASVs assigned to *Viridiplantae*. After the removal of plant-associated ASVs, the remaining 2,625 ASVs were identified as fungal ASVs and used in the following analysis (see Table S6). Although we removed some ASVs that were assigned to plants, the sequencing depth was sufficiently high to capture most of the observed ASVs (see Fig. S1a and b).

The differences in alpha diversity indices determined by the Shannon index and richness were analyzed for both bacteria and fungi (Fig. 2a; see also Table S7). In general, the alpha-diversity of bacterial communities was higher than that of fungal communities. For Kelsey, the bacterial alpha-diversity increased gradually from the first stage to the

**TABLE 1** Temporal effects of sampling stage and *Gymnosporangium yamadae* infection on the structure of phyllosphere bacterial and fungal communities as determined by PERMANOVA

| Category | Bacterial community | | | | | | Fungal community | | | | | |
| | Kelsey | | | Flame | | | Kelsey | | | Flame | | |
| | F | $R^2$ | P | F | $R^2$ | P | F | $R^2$ | P | F | $R^2$ | P |
| --- | --- | --- | --- | --- | --- | --- | --- | --- | --- | --- | --- | --- |
| Sampling stage | 2.084 | 0.258 | 0.014 | 4.514 | 0.429 | 0.001 | 2.372 | 0.283 | 0.001 | 6.417 | 0.517 | 0.001 |
| *G. yamadae* infection | 1.213 | 0.034 | 0.276 | 1.675 | 0.047 | 0.144 | 3.127 | 0.084 | 0.014 | 2.491 | 0.068 | 0.042 |

fifth stage, with a slight decrease in the sixth stage in healthy leaves but a pronounced decrease in *G. yamadae*-infected leaves. The fungal alpha-diversity increased with the expansion of lesions in both healthy and *G. yamadae*-infected leaves. For Flame, the bacterial alpha-diversity decreased from the first stage to the sixth stage in both healthy and *G. yamadae*-infected leaves; however, the fungal alpha-diversity first decreased and then increased. Principal coordinate analysis (PCoA) and permutational multivariate analysis of variance (PERMANOVA) based on the weighted UniFrac distances of bacterial and fungal communities showed that bacterial or fungal communities differed significantly with the expansion of lesions (Table 1 and Fig. 2b). The variation in the microbial community was mainly explained by the sampling stage. For Kelsey, it explained 49.58 and 36.52% of the differences in the bacterial and fungal community structure, respectively; for Flame, it explained 41.64 and 41.23%, respectively. Leaf status (noninfected versus *G. yamadae* infected) was the second most important indicator for interpreting the variation in microbial communities. In addition, the beta dissimilarities of bacterial and fungal communities showed similar patterns across six sampling stages in both Kelsey and Flame, which showed a greater variation in the late stages (4 to 6) than in the early stages (1 to 3) (see Fig. S2).

**Changes in the taxonomic patterns of microbial community and differential microorganisms.** The composition of the bacterial community at the phylum and genus levels and the fungal community composition at the family level across the six stages of Kelsey and Flame are visualized in Fig. 3a (see also Fig. S3 in the supplemental material). In Kelsey and Flame, the dominant bacterial phyla were *Proteobacteria*, *Firmicutes*, *Actinobacteria*, and *Bacteroidetes*. At the genus level, most of the bacterial Kelsey ASVs were assigned to *Sphingomonas* (10%), *Methylobacterium* (7.4%), *Bacteroides* (5.8%), *Massilia* (5.5%), *Faecalibacterium* (5%), *Kineococcus* (4.3%), and *Pseudomonas* (3.8%). In Flame, bacterial ASVs were mainly classified into *Sphingomonas* (10.3%), *Methylobacterium* (5.5%), *Curtobacterium* (5.1%), *Bacteroides* (5%), *Faecalibacterium* (4.3%), *Massilia* (3.7%), *Kineococcus* (3%), and *Pseudomonas* (2.5%), respectively. At the family level, the four most dominant fungal groups in Kelsey were *Didymellaceae* (22.9%), *Pleosporaceae* (22.5%), *Cladosporiaceae* (13.3%), and *Mycosphaerellaceae* (8.0%). The majority of the fungal ASVs in Flame belonged to *Cladosporiaceae* (20.4%), *Pleosporaceae* (17.3%), *Mycosphaerellaceae* (10.5%) and *Venturiaceae* (9.6%), respectively.

Bipartite networks were used to illustrate the conserved and shared ASVs in the bacterial and fungal communities at different stages (Fig. 3b). We combined the first and second stages as the early stages, the third and fourth stages as the middle stages, and the fifth and sixth stages as the late stages. In summary, *G. yamadae*-infected leaves had a greater proportion of bacterial ASVs that were specific in the early stages and a greater proportion of fungal ASVs that were specific in the late stages compared to noninfected leaves. In other words, *G. yamadae*-infected leaves were enriched with more bacteria in the early stages and more fungi in the late stages.

To obtain an overview of the changes in the taxonomic composition of bacterial and fungal communities following *G. yamadae* infection and, as the lesions expanded, we performed differential expression analysis to identify differences in the microbial taxa (see Tables S8 and S9). In general, some microorganisms were gradually enriched from the first to the sixth stage, during which the relative abundances of some microorganisms decreased

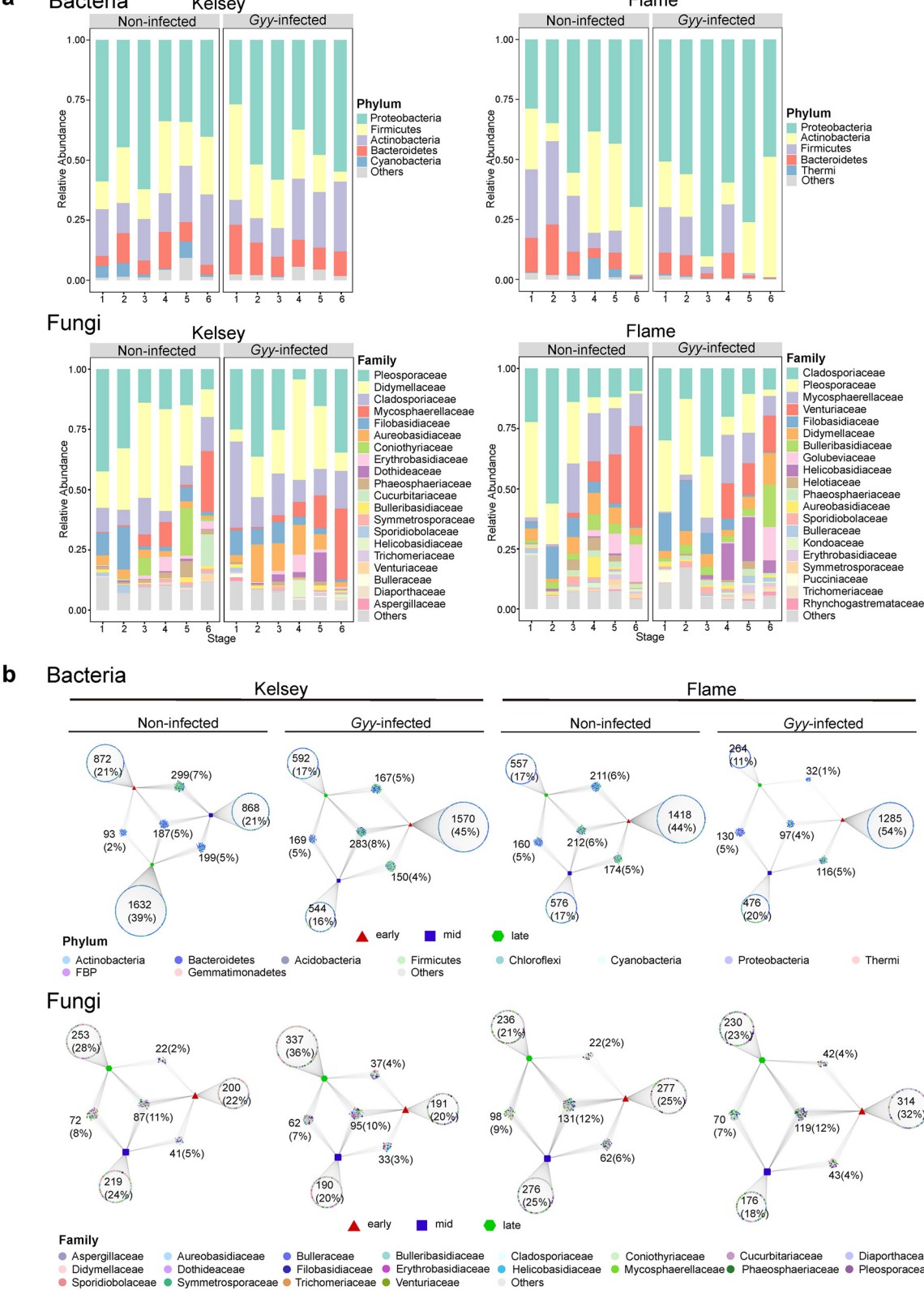

**FIG 3** (a) Taxonomic compositions of the bacterial community at the phylum level and fungal community at the family level inhabiting the phyllosphere of the two *Malus* cultivars (Flame and Kelsey). (b) Bipartite networks illustrating the specific and shared ASVs assigned to the early (1 and 2), middle (3 and 4), and late (5 and 6) stages. Each point represents one ASV. Different colors represent ASVs belonging to unique phyla (bacteria) or families (fungi).

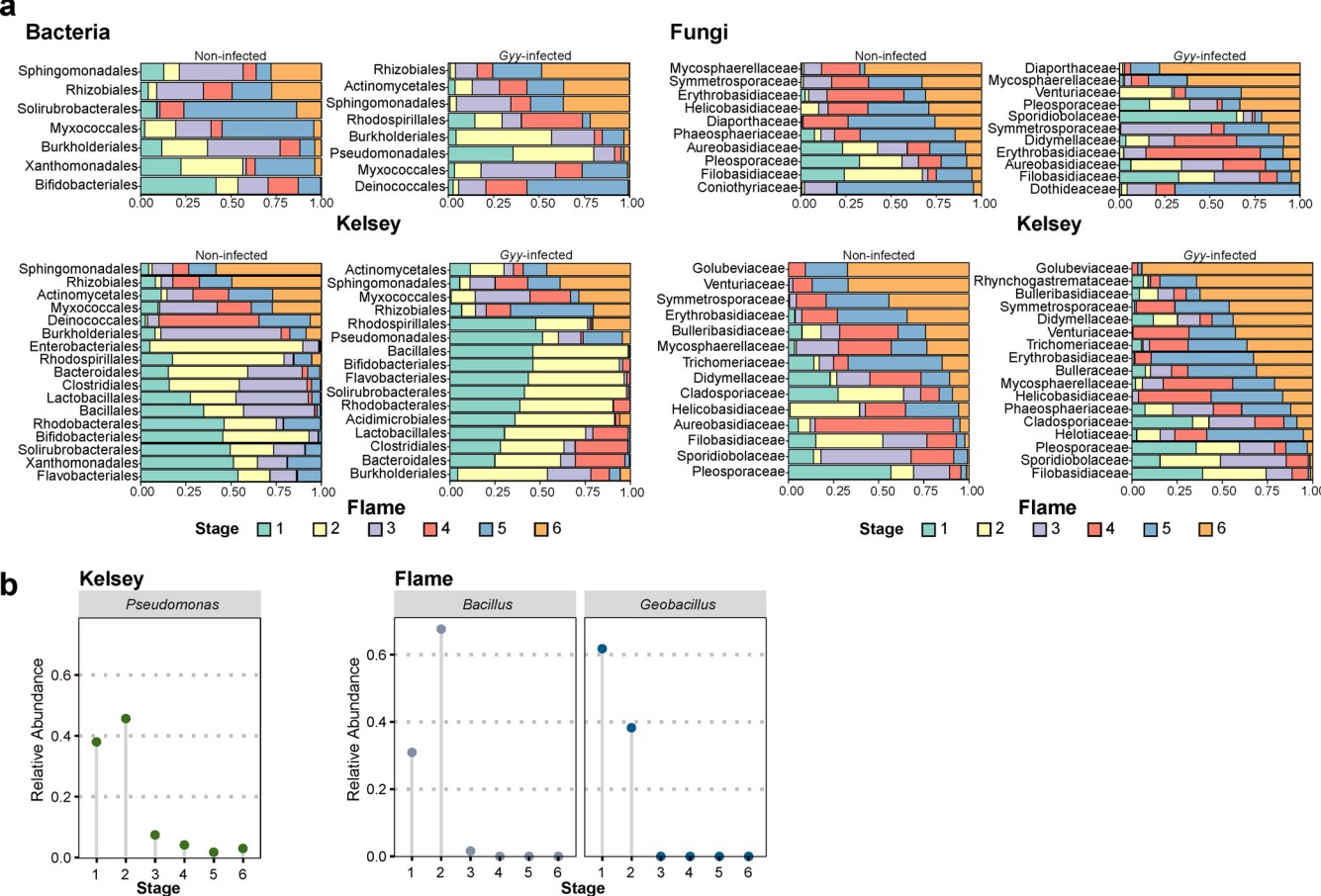

**FIG 4** (a) Changes in the abundance of differential bacterial orders and fungal families in both noninfected leaves and *G. yamadae*-infected leaves throughout lesion expansion in the two *Malus* cultivars (Flame and Kelsey). All abundances were normalized to a range from 0 to 1, with 1 representing the sum of each order. Differential analysis based on generalized linear model (GLM) was used to determine microbiota with significant differences across different developmental stages ($P < 0.05$, FDR corrected). (b) Visualization of dominant and distinctly variable genera in *Pseudomonadales Malus* Kelsey and the *Bacillales* of *Malus* Flame.

significantly. Specifically, in Kelsey, the relative abundance of seven and eight bacterial orders differed significantly with the enlargement of lesions in noninfected and *G. yamadae*-infected leaves, respectively (Fig. 4a). As lesions expanded, the relative abundances of *Deinococcales* and *Actinomycetales* significantly increased in *G. yamadae*-infected leaves, whereas the relative abundance of *Pseudomonadales* was significantly reduced. The relative abundances of these three orders did not change significantly in noninfected leaves. In Flame, the relative abundances of 17 and 16 bacterial orders differed significantly with the sampling stages in noninfected and *G. yamadae*-infected leaves, respectively (Fig. 4a). *Pseudomonadales* and *Acidimicrobiales* decreased significantly with the expansion of lesions in *G. yamadae*-infected leaves, whereas no significant changes were observed in noninfected leaves. Compared to noninfected leaves, *Clostridiales*, *Lactobacillales*, *Bacillales*, *Solirubrobacterales*, *Rhodobacterales*, *Flavobacteriales*, and *Solirubrobacterales* of bacterial communities were more significantly enriched in the early stages of *G. yamadae*-infected leaves.

We further identified the dominant and significantly changing genera in the order *Pseudomonadales* of Kelsey and in the order *Bacillales* of Flame, respectively (Fig. 4b). The results showed that the abundance of *Pseudomonas* was enhanced in early stages in Kelsey. In Flame, *Bacillus* and *Geobacillus* were the most abundant genera in early stages.

In Kelsey, the relative abundances of 10 and 11 fungal families differed significantly with the sampling stages in noninfected and *G. yamadae*-infected leaves, respectively

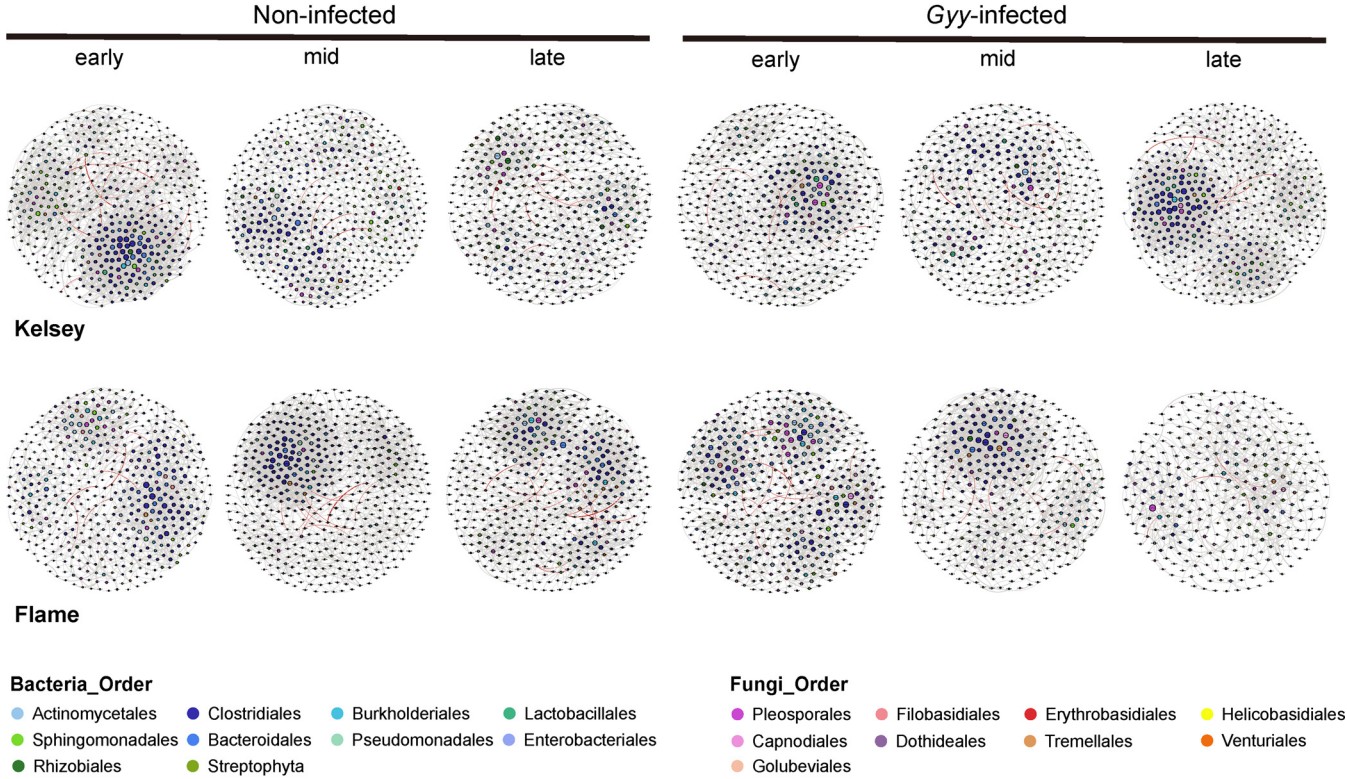

**FIG 5** Phyllosphere microbial co-occurrence networks of noninfected and *G. yamadae*-infected leaves of two *Malus* cultivars (Flame and Kelsey) at different stages (i.e., early, middle, and late stages). Positive correlations are shown in gray and negative correlations are displayed in red. The nodes are colored according to taxonomical classification of microbes (at the order level). The size of the node is proportional to the degree level.

(Fig. 4a). In *G. yamadae*-infected leaves, the relative abundances of *Venturiaceae*, *Didymellaceae*, and *Dothideaceae* increased significantly with lesion expansion, whereas the relative abundance of *Sporidiobolaceae* significantly decreased with no significant changes in noninfected leaves. In Flame, the relative abundances of 14 and 17 fungal families differed significantly with the sampling stages in noninfected and *G. yamadae*-infected leaves, respectively (Fig. 4a). The relative abundances of *Rhynchogastremataceae* and *Helotiaceae* significantly increased in *G. yamadae*-infected leaves with the lesion expansion and did not change significantly in noninfected leaves. Remarkably, *Didymellaceae* significantly increased with the expansion of lesions in the *G. yamadae*-infected leaves but decreased in the noninfected leaves. Phyllosphere fungi of the families *Golubeviaceae*, *Bulleribasidiaceae*, *Didymellaceae*, and *Trichomeriaceae* were more significantly enriched in the sixth stage of *G. yamadae*-infected leaves compared to noninfected leaves.

**Phyllosphere bacterial and fungal co-occurrence network complexity shifted with lesion expansion in both noninfected and *G. yamadae*-infected leaves.** To assess the impact of rust spot expansion on microbial interactions in noninfected and *G. yamadae*-infected leaves, we further performed microbial co-occurrence network analysis for Kelsey and Flame (Fig. 5). In Kelsey, the indices (clustering coefficient, network density, average degree, number of nodes, and number of edges) commonly used to assess the microbial network complexity first decreased and then increased in *G. yamadae*-infected leaves with lesion expansion. In the early and middle stages, these indices decreased in *G. yamadae*-infected leaves compared to noninfected leaves but increased in the late stages. However, modularization showed the opposite trend in the late stages. Bacterial orders accounted for a larger proportion of the network, and thus the trends of bacterial nodes and edges linking bacteria and bacteria were consistent with the trends of total nodes and edges (see Fig. S4a). In Flame, the number of nodes and edges as well as the clustering coefficient decreased in the *G. yamadae*-infected phyllosphere. Compared to noninfected leaves, the clustering

coefficient, the average degree, and the number of edges in *G. yamadae*-infected leaves decreased in the mid and late stages, but increased in the early stages. Modularization increased in *G. yamadae*-infected leaves compared to that in noninfected leaves (see Fig. S4b).

**Metabolites-induced potential recruitment of phyllosphere microbiota in two crabapple cultivars.** A correlation heatmap analysis was conducted between the selected metabolites and phyllosphere microbial taxa that were significantly different during rust spot expansion to reveal the pivotal roles of leaf metabolites in influencing phyllosphere microbiome assembly following *G. yamadae* infection (Fig. 6).

For Kelsey, amino acids, nucleotide sugars, diarylheptanoids, and carboxylic acids with aromatic rings were positively related to the microbes belonging to the bacterial order Pseudomonadales and the fungal family *Sporidiobolaceae*. Some aromatic compounds, flavonoids, lignin precursors, terpenoids, and quaternary ammonium salts were positively correlated with microbes in the bacterial order *Actinomycetales* and the fungal family *Venturiaceae*.

For Flame, peptides, nucleotides, diarylheptanoids, and carboxylic acids with aromatic rings positively correlated with the microbes in the bacterial orders *Acidimicrobiales*, *Lactobacillales*, *Bacillales*, *Rhodobacterales*, *Flavobacteriales*, and *Solirubrobacterales*. Some aromatic compounds, flavonoids, lignin precursors, terpenoids, and quaternary ammonium salts were positively related to microbes in the fungal families *Rhynchogastremataceae*, *Golubeviaceae*, and *Bulleribasidiaceae*.

## DISCUSSION

Plants are closely associated with numerous microorganisms in the rhizosphere, endosphere, and phyllosphere. Growing evidence shows that plants experiencing biotic or abiotic stress use a range of chemical stimuli to recruit health-promoting microbes and enhance their ability to mitigate stress, which is known as the "cry for help" strategy (8). Recent studies have extensively focused on the response of the rhizosphere microbiome (underground part) (44–46), whereas little is known about the phyllosphere microbiome (aboveground part). In this study, we profiled the dynamics of phyllosphere fungal and bacterial communities and key leaf metabolites in two *Malus* cultivars (*Malus* spp. Flame and Kelsey) at six stages after infection with *G. yamadae* to address the fundamental mechanisms underlying the regulation of rust-induced metabolic profiles in the phyllosphere microbiome. Our findings advance our collective knowledge about the essential roles of leaf metabolites in regulating the phyllosphere microbiome during foliar pathogen infection and provide evidence for the hypothesis that host plants can evolve a "cry for help" strategy.

**Significant alterations in leaf metabolites occurred during rust-caused leaf lesion expansion.** When exposed to biotic or abiotic stress, plant tissues and organs often exhibit visible local symptoms, such as programmed death (47) and thickening of the cuticle (27). In addition, spotted coloring in plants is also an important symptom, indicating a visible defense against infection (48). We have documented a significant positive correlation between flavonoid and lesion enlargement. An increase in flavonoid content coincided with spot expansion also in rust-infected apple leaves (49). In addition, other subclasses of secondary metabolites (e.g., some coumarins, terpenoids, aromatic compounds, quaternary ammonium salts, lignin, and lignin precursors) also increased significantly with an expansion of rust lesions in crabapple leaves. These compounds are widely known for their antimicrobial and antioxidant properties (34, 50). For example, coumarins and terpenoids accumulate around infection sites, inhibit pathogen growth, or cause plant defense responses (12, 51). Various aromatic compounds are released in the form of signaling molecules and carbon sources for heterotrophic organisms to fight pathogens (52). Some quaternary ammonium salts are produced in response to various environmental stresses to protect cell membranes, regulate enzymes, and detoxify reactive oxygen species (53). Increased levels of lignin and its precursors support the reinforcement of cell walls following pathogen infection (54, 55). In the present study, we also observed that amino acids, peptides, nucleotides, nucleotide sugars, diarylheptanoids, and carboxylic acids with aromatic rings

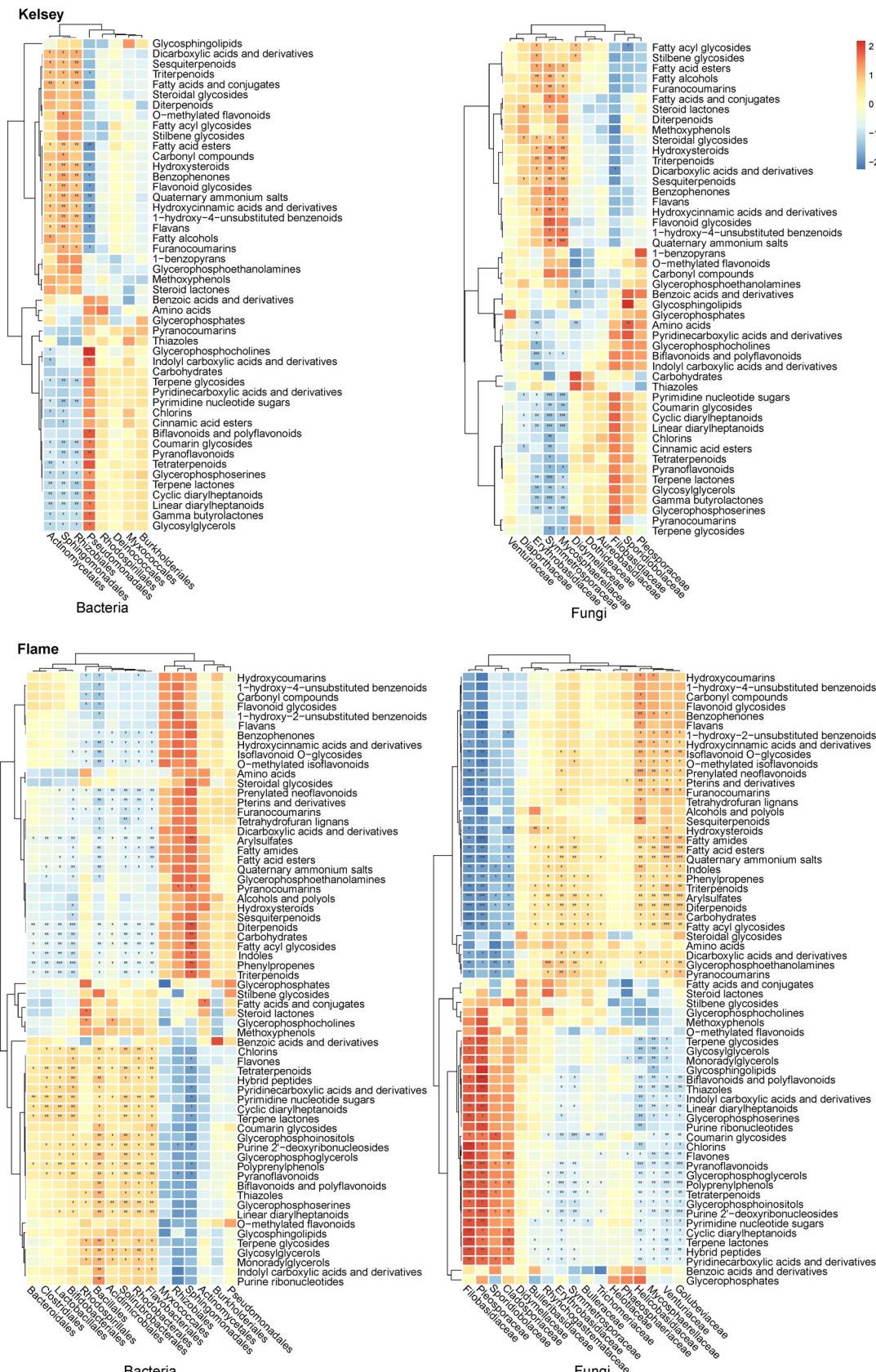

**FIG 6** Spearman's correlation heatmap analysis between the differential microorganisms and the differential metabolites with a *P* value of <0.05 (after FDR correction) in Kelsey and Flame.

were present at higher concentrations in the early stages than in other stages. Nucleotide sugars, intricate components of the cell wall, are essential in repairing penetrating lesions caused by pathogens (56). Amino acids and long-chain organic acids can supply nutrients to microorganisms (15, 57). Therefore, we speculate that the accumulation of various primary metabolites in the early stages was intended to supply nutrients and energy to specific microorganisms or to restore damaged cell walls. In addition, the high enrichment of secondary metabolites was supposed to defend against the initial rust infection.

**Phyllosphere microbial community diversity and structural changes with rust lesion expansion.** In recent years, several publications have reported that pathogen infections may result in changes to the microbiome in many pathosystems (33, 58, 59). In this study, we observed a higher alpha diversity of bacterial communities than that of fungal communities, and similarly, an elevated bacterial alpha diversity was observed in *Arabidopsis* root and soybean rhizosphere systems in previous studies (60, 61). Notably, we observed that the alpha diversity of both bacterial and fungal communities in healthy Kelsey leaves increased progressively from the first to the fifth stage, with a slight decrease in the sixth stage; this suggests a potential mechanism by which the highly diverse microbial communities in healthy leaves can trigger increasingly intensive interactions with the plant immune systems to avoid pathogens (62). Conversely, in *G. yamadae*-infected leaves, bacterial and fungal communities exhibited opposite responses toward rust spot expansion, with an initial decrease in alpha diversity in the early stages and a subsequent increase. We postulate that the roles of bacterial and fungal communities may change at different stages and that these changes may have resulted from different strategies used by different microbial kingdoms to combat rust infection (63). We also speculate that increased alpha diversity in the fungal community may be induced by pathogen invasion, which suppresses host defenses, allows rapid fungal colonization and infection, and consequently results in bacterial communities being at an inferior condition in nutrient competition. The PCoA results support our expectation that microbial community assemblages were mainly influenced by the developmental stage of rust, with microbial communities of healthy and *G. yamadae*-infected leaves being significantly different among stages. In support of this finding, Xiong et al. (63) reported that plant's developmental stage could explain the largest variations in bacterial and fungal communities.

**Shift in composition of phyllosphere microbial communities with *G. yamadae* infection.** The taxonomic composition of the phyllosphere bacterial community revealed that *Proteobacteria*, *Firmicutes*, *Actinobacteria*, and *Bacteroidetes* were dominant in the two *Malus* cultivars, although the relative abundance differed between healthy and *G. yamadae*-infected leaves. This result confirms that the phyllosphere bacterial community is mainly composed of *Proteobacteria*, *Bacteroidetes*, *Actinobacteria*, and *Firmicutes*, with *Proteobacteria* members accounting for approximately 50% of the community composition (5). In addition, the taxonomic composition of fungal communities showed that the major families (e.g., *Pleosporaceae*, *Cladosporiaceae*, *Mycosphaerellaceae*, and *Filobasidiaceae*) belonged to Ascomycota and Basidiomycota, which are the most abundant phyla of phyllosphere fungal communities observed in previous studies (64, 65).

Some fungal species belonging to *Cladosporium* group have, reportedly, actively protected plants from various biotic stresses. For example, an inoculation experiment demonstrated that a *Cladosporium* species could reduce the severity of poplar rust as a candidate antagonist (66). A *Rhizoctonia solani* infection can shape the rhizosphere microbial community and specifically accumulate beneficial *Pseudomonas* (46). In accordance with these findings, our results demonstrated that the relative abundances of *Cladosporiaceae* and *Pseudomonas* increased significantly in *G. yamadae*-infected leaves compared to healthy leaves regardless of cultivar type. Thus, we can reasonably speculate that the increase in *Cladosporiaceae* and *Pseudomonas* in *G. yamadae*-infected leaves may help promote the potential ability of their host plants to resist rust. Notably, the taxonomic compositions of fungal communities colonizing the phyllosphere of the two *Malus* cultivars differed, with specific families enriched. Cultivar-specific microbial composition may be related to differences in phenotype and immunity to variable microbes (62, 67).

Several bacteria are involved in host-pathogen interactions, such as *Bacillus subtilis* (68) and *Pseudomonas fluorescens* (69) isolated from rhizosphere or phyllosphere communities. These species can activate induced systemic resistance (ISR), thus promoting plant health. In the present study, the *Pseudomonadales* group was significantly enriched in the early stages of *G. yamadae*-infected leaves of the two *Malus* cultivars. Furthermore, its relative abundance decreased progressively with lesion enlargement, whereas no such changes were found in healthy leaves. Notably, the dominant genus *Pseudomonas* of Kelsey was enriched in the early stages. Consistently, members of *Pseudomonas* have been reported as being enriched in the root endosphere when attacked by the fungal pathogen *Rhizoctonia solani* (45). In our study, *Bacillus* and *Geobacillus*, the two dominant genera belonging to *Bacillales*, were highly abundant in the early stages of Flame. Furthermore, *Bacillales* were more significantly enriched in the early stages of *G. yamadae*-infected leaves than in noninfected leaves. This finding is consistent with the observations in previous studies where *Bacillus* is significantly enriched in diseased plants and is responsible for activating the plant immune system (15, 70). Moreover, the abundance of *Geobacillus* is higher when scab severity is lower in soil (71). These results suggest that plants may enhance the enrichment of specific microbial groups in the face of pathogen infection, as reported in previous studies (35, 66). These enriched microorganisms may serve as antagonistic candidates for *Malus* rust. Here, further culture-based experiments are required to determine whether these microorganisms exhibit antagonistic properties and their contribution to host-rust interactions.

For the phyllosphere fungal community, we found that *Diaporthaceae* and *Didymellaceae* were significantly enriched in the sixth stage of *G. yamadae*-infected leaves. *Diaporthe* and *Didymella* are major pathogenic members of the families *Diaporthaceae* and *Didymellaceae*, respectively, which have been significantly enriched in diseased plant roots in previous studies (61, 70). Therefore, these fungal groups may be opportunistic pathogens that colonize highly vulnerable diseased plants (61).

**Changes in co-occurrence of phyllosphere microbial community with the enlargement of lesions.** Co-occurrence network analysis has been widely applied to reveal complex associations within microbial communities (72). The connectivity and complexity in diseased host plants are commonly higher than those in the control (33, 73). Our study reveals that the phyllosphere microbiota of Kelsey had more complex association patterns in the late stages than those of noninfected leaves. In addition, modularization was lower in *G. yamadae*-infected leaves than in noninfected leaves at the late stages. The modules were interpreted as niches (74). Accordingly, the increased network complexity and overlapping ecological niches in the late stages may stimulate competition among microorganisms, consequently inhibiting pathogen colonization and infection. Complex networks can also reduce the success of pathogen invasion and be instrumental in supporting ecosystem function (75, 76).

However, there is also evidence that higher scab severity may lead to lower microbial co-occurrence network complexity (59); this is in accordance with our result that the connectedness among microorganisms in *G. yamadae*-infected leaves of Flame decreases as lesions expand. The reduction in network complexity may be attributable to the reduction in community organization, which destabilizes the network and results in weaker interactions among phyllosphere microorganisms (70, 77). The network complexity and modularization of *G. yamadae*-infected leaves were higher than those of noninfected leaves in the early stages. Larger modularity values may be associated with higher resource availability (78). According to our hypothesis, higher resource availability in the early stages and various microorganisms that increase the network complexity may allow plants to resist pathogens. This inconsistency in the two *Malus* cultivars may have resulted from differences in host genotypes, which induce distinct associations with phyllosphere microorganisms in response to rust infection.

**Leaf metabolites possibly modulate specific microbes.** Previous studies have thoroughly documented that secondary metabolites secreted by plants, such as coumarin, triterpenes, and flavonoids, are essential in fine-tuning the composition and function of the

plant microbiome (12, 13, 19, 20, 23, 79). Most of these studies on the effect of plant metabolites on plant-associated microbes have focused on belowground systems (13, 19, 80). The relative abundances of *Proteobacteria* (12) and *Pseudomonas* (19) around plant roots may be reduced owing to host-plant-secreted coumarin accumulation. Furthermore, terpene, aromatic compounds, and amino acids have reportedly modulated *Pseudomonas* groups in several pathosystems (16, 81, 82). Based on the research on phyllosphere plant-associated microbes, we found that aromatic compounds, flavonoids, furanocoumarins, terpenoids, and quaternary ammonium salts enriched with the enlargement of lesions negatively regulated *Pseudomonadales* in Kelsey. Amino acids, peptides, nucleotide sugars, diarylheptanoids, and carboxylic acids with aromatic rings were more abundant in the early stages of rust infection, which positively regulated the bacterial orders *Pseudomonadales* (in Kelsey) and *Bacillales* (in Flame).

Plant metabolites, as regulators of *Bacillus*, are commonly found in belowground systems research. For instance, the root system of *A. thaliana* can secrete large amounts of malic acid and recruit *Bacillus subtilis* FB17 after pathogen infection (15). Plant roots infected by *Fusarium oxysporum* increased tryptophan but reduced raffinose exudation, which enhanced root colonization by the beneficial bacterium *Bacillus amyloliquefaciens* SQR9 (17). *Pseudomonas* and *Bacillus* reportedly have the ability to produce microbial volatile organic compounds that directly inhibit plant pathogens by altering the transcriptional expression levels of several genes involved in motility and pathogenicity or indirectly by enhancing induced systemic resistance in plants (83). These results suggest that under biotic stresses, plants can recruit beneficial microbes by altering the synthesis and secretion of specific root exudates to enhance their stress tolerance (5, 15, 84, 85). Therefore, we speculate that the accumulation of various primary metabolites in the early stages of *G. yamadae*-infected leaves was intended to recruit beneficial *Pseudomonadales* and *Bacillales* and enhance host stress tolerance.

Plant metabolites also actively regulate fungi but mostly focus on the interactions between crop roots and their symbiotic fungi (86, 87). Our study found that some secondary metabolites that were enriched with the rust disease positively regulated fungi of the families *Rhynchogastremataceae* and *Golubeviaceae*. Currently, the roles of these fungi remain unclear. The large enrichment of secondary metabolites may have been intended to protect against the initial rust infection, and it may be the preferred carbon source for these fungi.

The key microbes enriched significantly following lesion expansion and regulated by metabolites are found tentatively in this study. However, whether these microbes have antagonistic features needs to be verified in the laboratory by following culture-based experiments.

## MATERIALS AND METHODS

**Sample collection.** From June to September 2021, a time course collection of noninfected and *G. yamadae*-infected leaves of Flame and Kelsey was performed at Tangjiabao Village, Yanqing District, Beijing (latitude, 40.5°N; longitude, 116°E) and at the south gate of Olympic Park, Beijing (latitude, 40°N; longitude, 116.38°E), respectively. Leaf samplings were collected at six stages, from the production of spermatogonia to the maturity of aecia. At each sampling site, three biological replicates of noninfected leaves and *G. yamadae*-infected leaves with relatively uniform leaf ages were collected simultaneously from a single tree to ensure consistency of genetic background and were subjected to microbiome analysis. Rust-diseased leaves from six stages were collected from one tree as biological replicates for further metabolic analysis. After rust diseased leaves were photographed in the field, all collected leaves were transported to the laboratory on dry ice and stored at −80°C before DNA and metabolite extraction.

**DNA extraction, library preparation, and sequencing.** For DNA extraction, 10 to 15 g of *Malus* leaves with rust symptoms (spermatogonia or aecia) and noninfected leaves were cut using sterile scissors and subjected to DNA extraction using the FastDNA SPIN kit for soil (MP Biomedicals, USA), according to the manufacturer's instructions. The DNA concentration and purity were determined using a NanoDrop 2000 UV-vis spectrophotometer (Thermo Scientific, Wilmington, DE) and electrophoresis on 1% agarose gels. The primer pairs 799F (AACMGGATTAGATACCCKG)/1193R (ACGTCATCCCCACCTTCC) (88) and ITS3F (GCATCGATGAAGAACGCAGC)/ITS4R (TCCTCCGCTTATTGATATGC) (89) were used to amplify the bacterial V5-V7 region of the 16S rRNA gene using nested PCR and amplify the internal transcribed spacer 2 (ITS2) region of the fungal ITS gene, respectively. Each sample was amplified in triplicate in a 20-$\mu$L reaction system that contained 5× TransStart FastPfu buffer (4 $\mu$L), 2.5 mM dNTPs (2 $\mu$L), a forward primer (5 $\mu$M, 0.8 $\mu$L), a reverse primer (5 $\mu$M, 0.8 $\mu$L), TransStart FastPfu DNA polymerase (0.4 $\mu$L), template

DNA (10 ng), and ddH$_2$O (up to 20 $\mu$L). The PCR amplification conditions were as follows: initial denaturation at 95℃ for 3 min, followed by 27 cycles of denaturation at 95℃ for 30 s, annealing at 55℃ for 30 s, extension at 72℃ for 45 s; a single extension at 72℃ for 10 min; and finishing at 10℃. All samples were amplified in triplicate for 13 cycles under identical conditions to those of the first-round PCR. The PCR product was extracted from a 2% agarose gel, purified using the AxyPrep DNA gel extraction kit (Axygen Biosciences, Union City, CA) according to the manufacturer's instructions, and quantified using a Quantus Fluorometer (Promega, USA). Purified amplicons were pooled in equimolar amounts and paired-end sequenced on an Illumina MiSeq PE300 platform (Illumina, San Diego, CA) according to the standard protocols of Majorbio Bio-Pharm Technology Co., Ltd. (Shanghai, China).

**Bioinformatics and statistical analysis.** All Illumina paired-end raw reads were processed using Quantitative Insights into Microbial Ecology 2 (QIIME2) and its plugins (90). To obtain ASVs, quality trimming, denoising, merging, and chimera detection were performed using the DADA2 plugin (91). Taxonomy for bacteria and fungi was assigned using the SILVA and UNITE reference databases, respectively. Alpha-diversity indices (Richness and Shannon index) and beta-diversity metrics (weighted UniFrac distance) were calculated using the QIIME2 diversity plugin.

For both *Malus* cultivars, the differences in alpha diversity of bacterial or fungal communities from non-infected leaves and *G. yamadae*-infected leaves were tested using one-way ANOVA and Tukey's HSD parametric tests utilizing the multcomp package (92). The weighted Unifrac distance matrices were computed and visualized using PCoA to investigate beta-diversity under different stages and leaf infection conditions. PERMANOVA was used to estimate the significance of differences in community composition among sample groups using the vegan package (93). Venn network diagrams were generated in CYTOSCAPE (94) using ASVs as source nodes and stages as target nodes. In this instance, the edges indicated the associations between the ASVs and stages. The differentially abundant ASVs for both bacteria and fungi were obtained using the generalized linear model (GLM) approach in the DESeq2 package (95). Every two sampling stages were compared in pairs ($P <$ 0.05, false discovery rate [FDR] corrected). Differential microorganisms occurring in two or more pairs were deemed responsive to rust infection and were used for visualization.

Co-occurrence network analysis was performed using the igraph package (96) to determine the effects of *G. yamadae* infection and sampling stage on phyllosphere microbial associations for both *Malus* cultivars. Data filtering was performed before constructing the co-occurrence network. Only ASVs with high relative abundances (top 10%) were kept to mitigate random variances (97). Statistically significant correlations ($P <$ 0.05, $r >$ 0.6) among ASVs were imported into the interactive platform Gephi (98, 99). We used a set of parameters, including the number of edges and nodes, average degree, and network density, to estimate the topology of the co-occurrence networks.

**Extraction and profiling of metabolites and statistical analysis.** The *G. yamadae*-infected leaves were ground to a powder in a frozen tissue grinder. Powder (50 mg) was added to 400 $\mu$L of methanol-water (4:1, vol/vol), and L-2-Cl-Phe (0.02 mg/mL) was used as an internal standard. After 30 min of ultra-sonication, the samples were incubated at $-20$℃ for 30 min. The extracts were then centrifuged at 13,000 rpm for 15 min at 4℃. Finally, the supernatant was analyzed using a Q-Exactive HF-X mass spectrometer equipped with an electrospray interface (Waters Corporation, Milford, MA), as described by Su et al. (100). Quality control (QC) samples were prepared by mixing equal volumes of all samples. A QC sample was inserted every 10 analytical samples to assess the stability of the analytical system and data reliability.

The raw data were preprocessed on the Majorbio platform and subsequently analyzed in R (101). Briefly, the preprocessing process included filtering of low-quality peaks, missing value padding, normalization, and relative standard deviation evaluations of QC samples. PCA was used to visualize the overall differences in the metabolomes of the six stages using the ropls package (101). Spearman's correlation analysis was performed to obtain a global overview of how the metabolite profile changes with lesion enlargement. Orthogonal partial least square-discriminant analysis (OPLS-DA) was used to identify differentially abundant metabolites based on their variable importance in the projection (VIP) values, which were evaluated by 7-fold cross-validation and response permutation testing (102). Wilcoxon's test analysis and fold change (FC) were calculated using the stats package (103). The metabolites that satisfied VIP $>1$, $P <$ 0.05 (after FDR correction) and FC $\geq$ 2 or $\leq$ 0.5 simultaneously were identified as differentially abundant between sampling stages (104). Finally, Spearman correlation analysis was performed based on the differential microorganisms and differential metabolites with a $P$ value of $<$0.05 (after FDR correction) and subsequently visualized using the pheatmap package (105) in R.

**Data availability.** The fungal and bacterial raw DNA sequences used in this study have been deposited in the NCBI Sequence Read Archive under accession numbers SRR21307697-SRR21307768.

## SUPPLEMENTAL MATERIAL

Supplemental material is available online only.
**SUPPLEMENTAL FILE 1**, XLSX file, 1.3 MB.
**SUPPLEMENTAL FILE 2**, XLSX file, 1.5 MB.
**SUPPLEMENTAL FILE 3**, XLSX file, 1.5 MB.
**SUPPLEMENTAL FILE 4**, XLSX file, 0.02 MB.
**SUPPLEMENTAL FILE 5**, XLSX file, 2.2 MB.
**SUPPLEMENTAL FILE 6**, XLSX file, 0.6 MB.
**SUPPLEMENTAL FILE 7**, XLSX file, 0.02 MB.

**SUPPLEMENTAL FILE 8**, XLSX file, 0.03 MB.
**SUPPLEMENTAL FILE 9**, XLSX file, 0.03 MB.
**SUPPLEMENTAL FILE 10**, PDF file, 0.4 MB.

## ACKNOWLEDGMENTS

This study was supported by the National Natural Science Foundation of China (grant 32101527), the Fundamental Research Funds for the Central Universities (grant 2021ZY02), and the China Postdoctoral Science Foundation (grant 2021M690418).

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
