## [Reviewer comments · Microbiology Spectrum]

Microbiology Spectrum

Metabolite-mediated responses of phyllosphere microbiota to rust infection in two *Malus* species

Yunxia Zhang, Bin Cao, Yumei Pan, Siqi Tao, and Naili Zhang

Corresponding Author(s): Naili Zhang, Beijing Forestry University

Review Timeline:

Submission Date:	September 22, 2022
Editorial Decision:	November 14, 2022
Revision Received:	February 14, 2023
Accepted:	February 24, 2023

Editor: Giuseppe Ianiri

Reviewer(s): The reviewers have opted to remain anonymous.

Transaction Report:

DOI: <https://doi.org/10.1128/spectrum.03831-22>

November 14, 2022

Dr. Naili Zhang
Beijing Forestry University
NO 35, Qinghua Eastern Road, Haidian District
Beijing China
China

Re: Spectrum03831-22 (Metabolites-mediated response of phyllosphere microbiota to rust infection in leaves of two *Malus* species)

Dear Dr. Naili Zhang:

Two reviewers evaluated your manuscript, with one recommending rejection. We believe the manuscript is interesting but the reviewers raised a number of concerns that need to be fully addressed by the authors before we can accept your manuscript. Failing to address the reviewers' concern will result in manuscript rejection.

Link Not Available

Sincerely,

Giuseppe Ianiri

Journals Department
Reviewer comments:

Reviewer #1 (Comments for the Author):

The phyllosphere is important for plant immunity, but, we know little about how pathogen infection affect metabolites and phyllosphere microbiota, especially for species beyond crops. Zhang et al. present a study of phyllosphere microbiota and metabolites in both symptomatic and asymptomatic leaves, and found some differences among them. Overall, the questions addressed in this study are interesting. However, I raised substantial concerns regarding the study, including the logic, reporting of statistics, description of methods, clarity of the text. Please see details below.

Major concern #1. Correlation does not imply causation. The authors explored the dynamics of phyllosphere fungal and bacterial communities at six stages respectively, in different sampling plots (line 497~509). And the authors concluded that metabolites enriched with the development of rust (line 31), "a trend towards a decreasing and then an increasing complexity of the microbial co-occurrence network" (line 24~25). However, we can not conclude metabolites and phyllosphere fungal and bacterial communities changed along six stages. In fact, the changes may be caused by the fact that plants with different metabolites or phyllosphere microbiota have different susceptibility to rust. In other words, it is unfair to compare those leaves that have not been infected with those diseased. Healthy leaves do not represent the state before infection, and they may never be infected. Inoculation experiment with continuous sampling are good methods to study the above questions, instead of one time sampling at the same place. The author should at least this alternative explanation in Discussion part in detail.

Major concern #2. The structure of the manuscript. To me, the biggest weakness of the Introduction part is that the introduction doesn't provide context to the reader for why we need to study the effects of rust infection on phyllosphere microbiota. In the first paragraph (line 51~63) of the Introduction, your scope was limited to two plants and one rust pathogen. In fact, your study should have a broader background, that is, how biological factors (pathogens) affect phyllosphere microbiota through host plant metabolites. Hence, it seems that the second paragraph (line 64~72) should be the beginning of the Introduction. While the background of target species can be placed at the very last of the Introduction, just before the hypothesis.

Major concern #3. Practical values. This manuscript described how rust pathogens affect microbiota through shifts in metabolites, instead of focusing on the control of rust. Hence, I suggest the authors weakening the part regarding biological control, e.g., line 45~46, line 58~63, line 493~495.

Line 45~46: How your study will contribute to the control of rust diseases? Please specify.

Line 58~63: In my view, this manuscript described how *Gymnosporangium yamadae* affected phyllosphere bacterial and fungal communities through shifts in metabolites, instead of focusing on how to control rust through microbiota or metabolites.

Line 76~77: The community evenness for what? Bacterial or fungal communities? Please specify.

Line 122: What is "LC-MS"?

Line 131~133: Moving to conclusion part, or just deleting this sentence?

Line 145: Maybe not "rust-induced metabolic alterations", just because the leaved in different metabolites also have different susceptibility to rust. See major concern #1.

Results: Please provide statistical parameters for each of your sentences in results part. E.g., F values with numerator and denominator degrees of freedom (df), and also p values for ANOVAs.

Line 547: Why not used Tukey's HSD test for multiple comparison?

Line 550: Principal Co-ordinates Analysis = PCoA; Principal Component Analysis = PCA.

Line 553: PERMANOVA was used to examine the difference in community composition among sample groups, rather than differences in beta-diversity.

Line 554: Generalized linear model. What is your link function and error distribution family? Please specify.

Line 555~556: What method was used for this comparison? Is Bonferroni correction was used?

Line 583: Whether the variables were standardized before PCA? Please specify.

Line 585: Not the metabolite profile changes with rust development, just differences in metabolite profile among different stages. See major concern #1.

Reviewer #2 (Comments for the Author):

In this study, Zhang et al. integrated microbiome and metabolomics data to systematically explore the dynamics of phyllosphere fungal and bacterial communities and key metabolites in two crabapple species (*Malus* 'Flame' and *M.* 'Coppurple') at six *Gymnosporangium yamadae* infection stages. By assessing the roles of leaf metabolites in the process of the phyllosphere microbiome formation of two *Malus* species, the authors identified several specific metabolites that have profoundly selective effects on the microbial community. Collectively, the newly identified link between leaf metabolites and phyllosphere microbiome

in this study will greatly contribute to the sustainable control of foliar diseases, however, it is difficult to publish this manuscript in *Microbiology Spectrum* in its current form and some additions and changes are needed to further strengthen the manuscript.

Major points:

1. In this manuscript, the authors did not introduce the genetic background of the two crabapples species (*Malus* 'Flame' and *M.* 'Coppurple') and their resistance to *Gymnosporangium yamadae*. In addition, whether the pathotypes of rust fungi from the two crabapples species are the same? Because different disease phenotypes were possibly attributed to different rust fungi with different virulence.
2. As the materials for this experiment were sampled, can the author ensure that the samples are only infected by *Gymnosporangium yamadae* and not infected by other pathogens. How to confirm that the changes of phyllosphere microorganisms are caused by the changes of metabolites induced by rust infection?
3. The bioinformatics analysis of the whole manuscript is too sketchy, and it is necessary that more in-depth analysis should be carried out.
4. When analyzing the Phyllosphere fungal community structure and diversity differed between different infection stages, I think it is necessary to separate the rust fungus for comparison, so as to determine whether the changes in the relative abundance of the rust fungus are consistent with the phenotype along with the infection process, and reveal complex associations within microbial communities between the rust fungus and other microorganisms during the Phyllosphere fungal co-occurrence network complexity shifted with rust development.
5. Some experiments should be done to test whether the identified metabolites were correlated with the host resistance.
6. In addition, this article has many problems with language expression. The manuscript would benefit from proofreading by a native English speaker.

Minor comments:

1. Line 272, "In contrast to non-infected leaves, several fungi from the family Golubeviaceae, Bulleribasidiaceae, Didymellaceae and Trichomeriaceae were more significantly enriched in the sixth stage of Gyy-infected leaves compared with these in the sixth stage of non-infected leaves." should be changed "Several fungi from the family Golubeviaceae, Bulleribasidiaceae, Didymellaceae and Trichomeriaceae were more significantly enriched in the sixth stage of Gyy-infected leaves compared with non-infected leaves."
2. Some common abbreviations have been highlighted in the text, and I think they should be mentioned again in the Figure legends. For example, "QC" should be indicated again in Figure 1 and Table S1.
3. According to the notes in Table S1, I do not know which stage of the sample AS_7 represents, and I think the sample of AS_2 may be missing.
4. In Table S4, Table S5 and Table S6, I think it should be indicated what each sample name represents respectively.
5. In Figure 2a, what do a, b and ab respectively mean? I think it needs to be marked. In addition, there is a difference between Shannon index and Richness in the sixth stage in "Coppurple" in figure, which I think should be explained.
6. Line 214, "The microbial community assembly was mainly influenced by the development stage of rust disease (bacteria: 'Coppurple' 49.58%, 'Flame' 41.64%; fungi: 'Coppurple' 36.52%, 'Flame' 41.23%), followed by rust infection (bacteria: 'Coppurple' 15.69%, 'Flame' 25%; fungi: 'Coppurple' 21.68%, 'Flame' 21.1%)." I think this sentence is ambiguous and not very clear.
7. In Figure 2b, I think the author should have marked the R2 and p values clearly instead of looking them in Table 1.
8. In the results of "Changes in microbial community composition and differential microorganisms after rust infection", I think that bacterial community composition is not very accurate at the order level and it should be analyzed at the phylum and genus levels.
9. Line 513, I do not think the "FastDNA SPIN Kit for Soil" is a good choice.
10. In Figure 4a, the meanings represented by the lines in different colors should be explained, and correlation analysis should be done. In addition, whether the data is modularized during the analysis.

Staff Comments:

Preparing Revision Guidelines

Please return the manuscript within 60 days; if you cannot complete the modification within this time period, please contact me. If you do not wish to modify the manuscript and prefer to submit it to another journal, please notify me of your decision immediately so that the manuscript may be formally withdrawn from consideration by Microbiology Spectrum.

The phyllosphere is important for plant immunity, but, we know little about how pathogen infection affect metabolites and phyllosphere microbiota, especially for species beyond crops. Zhang et al. present a study of phyllosphere microbiota and metabolites in both symptomatic and asymptomatic leaves, and found some differences among them. Overall, the questions addressed in this study are interesting. However, I raised substantial concerns regarding the study, including the logic, reporting of statistics, description of methods, clarity of the text. Please see details below.

Major concern #1. Correlation does not imply causation. The authors explored the dynamics of phyllosphere fungal and bacterial communities at six stages respectively, in different sampling plots (line 497~509). And the authors concluded that metabolites enriched with the development of rust (line 31), “a trend towards a decreasing and then an increasing complexity of the microbial co-occurrence network” (line 24~25). However, we can not conclude metabolites and phyllosphere fungal and bacterial communities changed along six stages. In fact, the changes may be caused by the fact that plants with different metabolites or phyllosphere microbiota have different susceptibility to rust. In other words, it is unfair to compare those leaves that have not been infected with those diseased. Healthy leaves do not represent the state before infection, and they may never be infected. Inoculation experiment with continuous sampling are good methods to study the above questions, instead of one time sampling at the same place. The author should at least this alternative explanation in Discussion part in detail.

Major concern #2. The structure of the manuscript. To me, the biggest weakness of the Introduction part is that the introduction doesn't provide context to the reader for why we need to study the effects of rust infection on phyllosphere microbiota. In the first paragraph (line 51~63) of the Introduction, your scope was limited to two plants and one rust pathogen. In fact, your study should have a broader background, that is, how biological factors (pathogens) affect phyllosphere microbiota through host plant metabolites. Hence, it seems that the second paragraph (line 64~72) should be the beginning of the Introduction. While the background of target species can be placed at the very last of the Introduction, just before the hypothesis.

Major concern #3. Practical values. This manuscript described how rust pathogens affect microbiota through shifts in metabolites, instead of focusing on the control of rust. Hence, I suggest the authors weakening the part regarding biological control, e.g., line 45~46, line 58~63, line 493~495.

Line 45~46: How your study will contribute to the control of rust diseases? Please specify.

Line 58~63: In my view, this manuscript described how *Gymnosporangium yamadai* affected phyllosphere bacterial and fungal communities through shifts in metabolites,

instead of focusing on how to control rust through microbiota or metabolites.

Line 76~77: The community evenness for what? Bacterial or fungal communities?
Please specify.

Line 122: What is “LC-MS”?

Line 131~133: Moving to conclusion part, or just deleting this sentence?

Line 145: Maybe not “rust-induced metabolic alterations”, just because the leaved in different metabolites also have different susceptibility to rust. See major concern #1.

Results: Please provide statistical parameters for each of your sentences in results part. E.g., F values with numerator and denominator degrees of freedom (df), and also p values for ANOVAs.

Line 547: Why not used Tukey’s HSD test for multiple comparison?

Line 550: Principal Co-ordinates Analysis = PCoA; Principal Component Analysis = PCA.

Line 553: PERMANOVA was used to examine the difference in community composition among sample groups, rather than differences in beta-diversity.

Line 554: Generalized linear model. What is your link function and error distribution family? Please specify.

Line 555~556: What method was used for this comparison? Is Bonferroni correction was used?

Line 583: Whether the variables were standardized before PCA? Please specify.

Line 585: Not the metabolite profile changes with rust development, just differences in metabolite profile among different stages. See major concern #1.

Reviewer #1 (Comments for the Author):

The phyllosphere is important for plant immunity, but, we know little about how pathogen infection affect metabolites and phyllosphere microbiota, especially for species beyond crops. Zhang et al. present a study of phyllosphere microbiota and metabolites in both symptomatic and asymptomatic leaves, and found some differences among them. Overall, the questions addressed in this study are interesting. However, I raised substantial concerns regarding the study, including the logic, reporting of statistics, description of methods, clarity of the text. Please see details below.

Major concern #1. Correlation does not imply causation. The authors explored the dynamics of phyllosphere fungal and bacterial communities at six stages respectively, in different sampling plots (line 497~509). And the authors concluded that metabolites enriched with the development of rust (line 31), “a trend towards a decreasing and then an increasing complexity of the microbial co-occurrence network” (line 24~25). However, we can not conclude metabolites and phyllosphere fungal and bacterial communities changed along six stages. In fact, the changes may be caused by the fact that plants with different metabolites or phyllosphere microbiota have different susceptibility to rust. In other words, it is unfair to compare those leaves that have not been infected with those diseased. Healthy leaves do not represent the state before infection, and they may never be infected. Inoculation experiment with continuous sampling are good methods to study the above questions, instead of one time sampling at the same place. The author should at least this alternative explanation in Discussion part in detail.

Reply: Thank you for your dedicated efforts on revision of this manuscript. We completely understand what you are saying. This is a major problem when collecting samples in the field because the initial infection of *Gymnosporangium yamadae* basidiospores on *Malus* leaves is asymptomatic. Therefore, we set the initial collection time to the formation of spermatogonia on leaves and ended with the mature of aecia. During sample collection, the infected leaves and healthy leaves with relatively uniform leaf age were synchronously collected on a single tree to ensure consistency of genetic background. In fact, our main task in this study is not to find whether leaves with different metabolites and microbiomes showed differential susceptibility to rust infection, but to focus mainly on the changes in leaf metabolites and phyllosphere microbial communities on rust-infected leaves as the lesions expanded and their potential regulatory relationships. As you mentioned, the inoculation experiment would be better because we can collect the inoculated leaf immediately after inoculating basidiospores, which is being done by other colleagues in our lab. We have modified this part in the discussion part.

Major concern #2. The structure of the manuscript. To me, the biggest weakness of the Introduction part is that the introduction doesn't provide context to the reader for why we need to study the effects of rust infection on phyllosphere microbiota. In the first paragraph (line 51~63) of the Introduction, your scope was limited to two plants and one rust pathogen. In fact, your study should have a broader background, that is,

how biological factors (pathogens) affect phyllosphere microbiota through host plant metabolites. Hence, it seems that the second paragraph (line 64~72) should be the beginning of the Introduction. While the background of target species can be placed at the very last of the Introduction, just before the hypothesis.

Reply: Indeed, a global overview of how pathogen-induced metabolites affect the plant microbiome should be placed at the front of Introduction. The two *Malus* species and *G. yamadae* should be considered as simply suitable materials to access our working hypotheses. We have rewritten the Introduction section according to your suggestions.

Major concern #3. Practical values. This manuscript described how rust pathogens affect microbiota through shifts in metabolites, instead of focusing on the control of rust. Hence, I suggest the authors weakening the part regarding biological control, e.g., line 45~46, line 58~63, line 493~495.

Reply: We thank the reviewer for this valuable comment. We have weakened the part regarding biological control. However, lines 493 to 495 imply that the large enrichment of secondary metabolites with rust development was originally intended to defend against initial rust infection and may be the preferred carbon source for fungi that were positively regulated by these metabolites. It does not involve the use of beneficial microbes and microbial metabolites for biological control of this disease.

Line 45~46: How your study will contribute to the control of rust diseases? Please specify.

Reply: When infected by pathogens, plants may directly suppress pathogens by producing specific metabolites (Stringlis et al., 2018; Seybold et al., 2020). Besides, they may fight against pathogen by recruiting potentially beneficial microbes through specifically enriched metabolites (Liu and Brettell, 2019). We identified metabolites that changed significantly with lesions expanded and found significant association of beneficial microorganisms with these metabolites, merely through the metabolic and microbiome analyses. Whether these microorganisms have antagonistic features and their contribution in host-rust interactions remains to be determined through culture-based experiments. This is what we plan to do next. We have modified this expression throughout the text and weakened the “disease control” part.

- Stringlis IA, Yu K, Feussner K, de Jonge R, Van Bentum S, Van Verk MC, Berendsen RL, Bakker PAHM, Feussner I, Pieterse CMJ. 2018. MYB72-dependent coumarin exudation shapes root microbiome assembly to promote plant health. *Proc Natl Acad Sci USA* 115:5213-5222.
- Seybold H, Demetrowitsch TJ, Hassani MA, Szymczak S, Reim E, Hauelsen J, Lübbers L, Rühlemann M, Franke A, Schwarz K, Stukenbrock EH. 2020. A fungal pathogen induces systemic susceptibility and systemic shifts in wheat metabolome and microbiome composition. *Nat Commun* 11:1910.
- Liu HW, Brettell LE. 2019. Plant Defense by VOC-Induced Microbial Priming. *Trends Plant Sci* 24:187-189.

Line 58~63: In my view, this manuscript described how *Gymnosporangium yamadae*

affected phyllosphere bacterial and fungal communities through shifts in metabolites, instead of focusing on how to control rust through microbiota or metabolites.

Reply: Indeed, our main objective was to figure out whether changes in leaf metabolites caused by *G. yamadae* infection could affect the phyllosphere microbial communities, and to screen some potentially beneficial microorganisms that can exert disease-suppressive effects to *G. yamadae* through bioinformatic methods. However, the functions of these candidate microbes need to be verified by culture-based experiments in the laboratory, and therefore we will not emphasize this part in the revised manuscript.

Line 76~77: The community evenness for what? Bacterial or fungal communities? Please specify.

Reply: That is the bacterial community evenness and community structure, and this information has been added in page 5, lines 93~94, highlighted in yellow.

Line 122: What is “LC-MS”?

Reply: The full name of “LC-MS” is “Liquid Chromatography-Mass Spectrometry”, we have added the full name in page 6, line 125, highlighted in yellow.

Line 131~133: Moving to conclusion part, or just deleting this sentence?

Reply: We have deleted this sentence.

Line 145: Maybe not “rust-induced metabolic alterations”, just because the leaved in different metabolites also have different susceptibility to rust. See major concern #1.

Reply: We agree with you. And we have modified this part. Please see page 7, lines 149~150.

Results: Please provide statistical parameters for each of your sentences in results part. E.g., F values with numerator and denominator degrees of freedom (df), and also p values for ANOVAs.

Reply: We have added F values with numerator and denominator degrees of freedom (df), and *p* values for one-way ANOVA in the supplementary file (Table S7). It should be noted that a set of data (highlighted in yellow) in Table S7 that cannot be analyzed using ANOVA because it does not meet the homogeneity of variance. Instead, it was analyzed using Kruskal-Wallis test.

Line 547: Why not used Tukey’s HSD test for multiple comparison?

Reply: We initially used the LSD test for two reasons. First, the LSD test is more sensitive and can capture small differences. Second, the LSD test may focus on reducing type II error. However, the disadvantage is to increase the likelihood of type I error. In addition, considering the equal sample size per group, we instead used the Tukey’s HSD test for multiple comparisons. Compared to the LSD test, it still validated our main findings although some significance results of the Tukey’s HSD

test changed. We have accordingly modified this part in the main text and the Figure 2a.

Line 550: Principal Co-ordinates Analysis = PCoA; Principal Component Analysis = PCA.

Reply: Thank you for your reminder, we have corrected this typo.

Line 553: PERMANOVA was used to examine the difference in community composition among sample groups, rather than differences in beta-diversity.

Reply: Indeed, the PERMANOVA was used to examine the difference in community composition among sample groups. We have modified the statement about it in page 26, lines 563~565, highlighted in yellow.

Line 554: Generalized linear model. What is your link function and error distribution family? Please specify.

Reply: The read count is proceeded with the GLM of the negative binomial family with a logarithmic link.

Line 555~556: What method was used for this comparison? Is Bonferroni correction was used?

Reply: We run the samples from all stages together, and then use the contrast argument of the 'results' function to extract comparisons of each two stages after fitting the model using DESeq2 (Love et al., 2014). We used false discovery rate (FDR) to correct the p value, and we have added this information in page 27, line 570, highlighted in yellow.

- Love MI, Huber W, Anders S. 2014. Moderated estimation of fold change and dispersion for RNA-seq data with DESeq2. *Genome Biol* 15:550.

Line 583: Whether the variables were standardized before PCA? Please specify.

Reply: We preprocessed the raw data by filtering low quality peaks, missing value padding, normalization and relative standard deviation (RSD) evaluation of quality control (QC) samples. In addition, the variables were standardized before PCA.

Line 585: Not the metabolite profile changes with rust development, just differences in metabolite profile among different stages. See major concern #1.

Reply: As we replied before, it could be more appropriate to say the metabolite profile changes with our collection stages. We have modified the relevant statements about it across the main text of this manuscript.

Reviewer #2 (Comments for the Author):

In this study, Zhang et al. integrated microbiome and metabolomics data to systematically explore the dynamics of phyllosphere fungal and bacterial communities and key metabolites in two crabapple species (*Malus* 'Flame' and *M.* 'Coppurple') at six *Gymnosporangium yamadae* infection stages. By assessing the roles of leaf metabolites in the process of the phyllosphere microbiome formation of two *Malus* species, the authors identified several specific metabolites that have profoundly selective effects on the microbial community. Collectively, the newly identified link between leaf metabolites and phyllosphere microbiome in this study will greatly contribute to the sustainable control of foliar diseases, however, it is difficult to publish this manuscript in *Microbiology Spectrum* in its current form and some additions and changes are needed to further strengthen the manuscript.

Reply: Many thanks for the poignant suggestions! We have completely revised the manuscript according to you and the other reviewer. Hope the improved version meets the requirement of *Microbiology Spectrum*.

Major points:

1. In this manuscript, the authors did not introduce the genetic background of the two crabapples species (*Malus* 'Flame' and *M.* 'Coppurple') and their resistance to *Gymnosporangium yamadae*. In addition, whether the pathotypes of rust fungi from the two crabapples species are the same? Because different disease phenotypes were possibly attributed to different rust fungi with different virulence.

Reply: First, the scientific name of the crabapple cultivar collected at the south gate of Olympic Park, Beijing (latitude 40°N, longitude 116.38°E) was incorrect in the original draft. The correct name should be *Malus* 'Kelsey' instead of *M.* 'Coppurple'. This mistake was mainly due to wrong information plate of the collected tree. It was not until we looked up the genetic information of the crabapple cultivars that we discovered, and we have double-checked and found the morphological traits of the sampled leaves should be *M.* 'Kelsey'. We apologize for this error and thank the reviewer for kind reminder, and we have corrected it throughout the text.

The two *Malus* crabapples used in our study are two important ornamental resources in China. Due to the large number of cultivars in the *Malus* group, their genetic background is still mostly lacking. In terms of pathogen, research on *G. yamadae* have been mainly limited to morphological observations over the past few decades (Cummins and Hiratsuka, 2003; Kern, 1973), the phylogenetic and transcriptomic studies at the molecular level have only started in recent years (Tao et al., 2017, 2019, 2020), the pathogen pathogenesis and *Malus* resistance have been studied so far not on crabapples. However, our main objective was to study the metabolite-mediated response patterns of the microbial communities, separately for the two crabapples, rather than to compare the differences between them. Therefore,

the lacking information of pathogen virulence and host resistance did not hinder us from exploring the metabolic-induced microbial community shifts. With this in mind, we have added the information on the parental genetics of *M. 'Flame'* and *M. 'Kelsey'* in the Introduction section (page 5, lines 103~105, highlighted in yellow).

- Cummins G, Hiratsuka Y. 2003. Illustrated genera of rust fungi. APS Press, St Paul, MN, U.S.A.
- Kern FD. 1973. A revised taxonomic account of *Gymnosporangium*. Pennsylvania State University Press.
- Tao SQ, Cao B, Tian CM, Liang YM. 2017. Comparative transcriptome analysis and identification of candidate effectors in two related rust species (*Gymnosporangium yamadae* and *Gymnosporangium asiaticum*). BMC Genomics 18:651.
- Tao SQ, Cao B, Morin E, Liang YM, Duplessis S. 2019. Comparative transcriptomics of *Gymnosporangium spp.* teliospores reveals a conserved genetic program at this specific stage of the rust fungal life cycle. BMC Genomics 20:723.
- Tao SQ, Auer L, Morin E, Liang YM, Duplessis S. 2020. Transcriptome analysis of apple leaves infected by the rust fungus *Gymnosporangium yamadae* at two sporulation stages. Mol Plant Microbe Interact 33:444-461.

2.As the materials for this experiment were sampled, can the author ensure that the samples are only infected by *Gymnosporangium yamadae* and not infected by other pathogens. How to confirm that the changes of phyllosphere microorganisms are caused by the changes of metabolites induced by rust infection?

Reply: Admittedly, it is inevitable that the leaves could be infected by pathogens other than rust fungi, but what we can confirm that the rust fungus is the dominant pathogen in the early stages because of the appearance of typical rust sporulation structures (spermatogonia and aecia) and that the distinct discolored spots on leaves is due to rust fungi infection, which was found in previous studies (Lu et al., 2017; Liu et al., 2019), and this particular discoloration reaction was found in our study (Tao et al., 2020). Therefore, we also sampled the disease spots strictly to ensure that the changes in metabolites followed the changes in the disease spots. Indeed, there may be other secondary pathogens in the later stages of rust infection (after the maturation of aeciospores), but this was not the focus of our attention, and therefore we did not isolate and identify these pathogens.

- Lu Y, Chen Q, Bu Y, Luo R, Hao S, Zhang J, Tian J, Yao Y. 2017. Flavonoid accumulation plays an important role in the rust resistance of *Malus* plant leaves. Front Plant Sci 8:1286.
- Liu PY, Wang YL, Meng JX, Zhang X, Zhou J, Han ML, Yang C, Gan LX, Li HH. 2019. Transcriptome sequencing and expression analysis of genes related to anthocyanin biosynthesis in leaves of *Malus 'Profusion'* infected by Japanese apple rust. Forests 10:665.
- Tao SQ, Auer L, Morin E, Liang YM, Duplessis S. 2020. Transcriptome analysis of apple leaves infected by the rust fungus *Gymnosporangium yamadae* at two sporulation stages. Mol Plant Microbe Interact 33:444-461.

3.The bioinformatics analysis of the whole manuscript is too sketchy, and it is necessary that more in-depth analysis should be carried out.

Reply: We have made in-depth analysis as you suggested. First, we added bacterial community composition at the phylum level and genus level. At the ASV level, we used Venn networks diagrams to describe the differences between the communities at different stages from bacterial and fungal samples. Second, we visualized the dominant and significantly altered genera of potentially beneficial bacteria. Finally, we re-performed the correlation analysis and visualization of the network and added the clustering coefficient and modularization indices.

4. When analyzing the Phyllosphere fungal community structure and diversity differed between different infection stages, I think it is necessary to separate the rust fungus for comparison, so as to determine whether the changes in the relative abundance of the rust fungus are consistent with the phenotype along with the infection process, and reveal complex associations within microbial communities between the rust fungus and other microorganisms during the Phyllosphere fungal co-occurrence network complexity shifted with rust development.

Reply: Indeed, it is informative to determine the relative abundance of *G. yamadae* with the infection process, however, it was difficult to extract the ASVs of *G. yamadae* from all phyllosphere fungal ASVs. Because the relative abundance of rust fungi (order Pucciniales) was quite low (0.2 % in the Fungal Unite Database that we used for taxonomical classification) in this study, not to mention that *G. yamadae*, an endemic species in Asia, for which the molecular data were not available until recent years (out of 171,458 ASVs, only 13 belong to *Gymnosporangium*). Therefore, the relative abundance of *G. yamadae* obtained by annotating in databases is quite low and poorly displayed by graphs. Since we only focus on the pattern of non-rust microbial changes with lesions expanded in this study, we did not retain the material to perform qPCR tests which could capture changes in abundance with relative accuracy.

5. Some experiments should be done to test whether the identified metabolites were correlated with the host resistance.

Reply: Yes, the metabolites that were identified in this study and had a significant regulatory effect in microbial community reconstruction will be further validated in the laboratory to demonstrate their inhibitory effect on rust. Some colleagues in our lab have already conducted follow-up experiments based on the results.

6. In addition, this article has many problems with language expression. The manuscript would benefit from proofreading by a native English speaker.

Reply: Thank you for the helpful suggestions. After carefully checking the manuscript by ourselves, a native English speaker help us polish the manuscript.

Minor comments:

1. Line 272, "In contrast to non-infected leaves, several fungi from the family Golubeviaceae, Bulleribasidiaceae, Didymellaceae and Trichomeriaceae were more significantly enriched in the sixth stage of Gyy-infected leaves compared with these in

the sixth stage of non-infected leaves." should be changed "Several fungi from the family Golubeviaceae, Bulleribasidiaceae, Didymellaceae and Trichomeriaceae were more significantly enriched in the sixth stage of Gyy-infected leaves compared with non-infected leaves."

Reply: Thanks to you for your good comments. We have revised this sentence according to the constructive suggestion (page 14, lines 283~286, highlighted in yellow).

2. Some common abbreviations have been highlighted in the text, and I think they should be mentioned again in the Figure legends. For example, "QC" should be indicated again in Figure 1 and Table S1.

Reply: We have indicated these abbreviations in the legends.

3. According to the notes in Table S1, I do not know which stage of the sample AS_7 represents, and I think the sample of AS_2 may be missing.

Reply: We sampled 'Kelsey' twice at the second stage of disease development (II). The first collected leaf was named AS_2, but we found fewer spots on the leaf that might be difficult to extract metabolites so we conducted a second sampling on the same day. Collected leaves with more spots were named AS_3 to distinguish it from AS_2. The later sampling was named from AS_4. We apologize for the misleading numbering, we have explained accordingly in the legends.

4. In Table S4, Table S5 and Table S6, I think it should be indicated what each sample name represents respectively.

Reply: Thank you for the helpful suggestions. We have explained what each sample name represents in the legends.

5. In Figure 2a, what do a, b and ab respectively mean? I think it needs to be marked. In addition, there is a difference between Shannon index and Richness in the sixth stage in "Coppurple" in figure, which I think should be explained.

Reply: We have indicated the meaning of the different letters in the legend of Figure 2. And sorry, we did not follow the question here: "there is a difference between Shannon index and Richness in the sixth stage in Coppurple", because we did not compare the two alpha diversity indices.

6. Line 214, "The microbial community assembly was mainly influenced by the development stage of rust disease (bacteria: 'Coppurple' 49.58%, 'Flame' 41.64%; fungi: 'Coppurple' 36.52%, 'Flame' 41.23%), followed by rust infection (bacteria: 'Coppurple' 15.69%, 'Flame' 25%; fungi: 'Coppurple' 21.68%, 'Flame' 21.1%)." I think this sentence is ambiguous and not very clear.

Reply: We have modified this sentence to "The variation in the microbial community was mainly explained by the sampling stage. For 'Kelsey', it explained 49.58% and 36.52% of the differences in the bacterial and fungal community structure, respectively; for 'Flame', it explained 41.64% and 41.23%, respectively. Leaf status

(non-infected vs. *Gyy*-infected) was the second most important indicator for interpreting the variation in microbial communities”, please see page 10, lines 213~217, highlighted in yellow.

7. In Figure 2b, I think the author should have marked the R^2 and p values clearly instead of looking them in Table 1.

Reply: We have marked the R^2 and p values in figure 2b as suggested.

8. In the results of "Changes in microbial community composition and differential microorganisms after rust infection", I think that bacterial community composition is not very accurate at the order level and it should be analyzed at the phylum and genus levels.

Reply: We have modified the analysis of bacterial community composition at the order level to phylum level and genus level. Please see figure 3a and figure S3.

9. Line 513, I do not think the "FastDNA SPIN Kit for Soil" is a good choice.

Reply: We will carefully consider your suggestion about DNA extraction kit in the following studies. In this study, we have done comparative experiments, and good results can be achieved by DNA extraction with FastDNA SPIN Kit for Soil. In addition, there are other studies using this kit to extract DNA from plant leaves (Li et al., 2022; Song et al., 2022).

- Li Y, Pan J, Zhang R, Wang J, Tian D, Niu S. 2022. Environmental factors, bacterial interactions and plant traits jointly regulate epiphytic bacterial community composition of two alpine grassland species. *Sci Total Environ.* 836:155665.
- Song M, Sun B, Li R, Qian Z, Bai Z, Zhuang X. 2022. Successions and interactions of phyllospheric microbiome in response to NH_3 exposure. *Sci Total Environ.* 837:155805.

10. In Figure 4a, the meanings represented by the lines in different colors should be explained, and correlation analysis should be done. In addition, whether the data is modularized during the analysis.

Reply: Thank you for the reminder, we neglected to filter the correlation matrix with adjusted p value when we did the network analysis before, so we modified this and re-visualized it, please see Figure 4. We explained what the different color lines represent, plus added the clustering coefficient and modularization indices of the network.

February 24, 2023

Dr. Naili Zhang
Beijing Forestry University
NO 35, Qinghua Eastern Road, Haidian District
Beijing China
China

Re: Spectrum03831-22R1 (Metabolite-mediated responses of phyllosphere microbiota to rust infection in two Malus species)

Dear Dr. Naili Zhang:

Your manuscript has been accepted, and I am forwarding it to the ASM Journals Department for publication. You will be notified when your proofs are ready to be viewed.

Sincerely,

Giuseppe Ianiri
Editor, Microbiology Spectrum
